# Sustainable Esterification of a Soda Lignin with Phloretic Acid

**DOI:** 10.3390/polym13040637

**Published:** 2021-02-21

**Authors:** Antoine Adjaoud, Reiner Dieden, Pierre Verge

**Affiliations:** 1Luxembourg Institute of Science and Technology, Materials Research and Technology Department, 5 Avenue des Hauts-Fourneaux, L-4362 Esch-sur-Alzette, Luxembourg; antoine.adjaoud@list.lu (A.A.); reiner.dieden@list.lu (R.D.); 2University of Luxembourg, 2, Avenue de l’Université, L-4365 Esch-sur-Alzette, Luxembourg

**Keywords:** lignin, sustainable, esterification, solubility

## Abstract

In this work, a sustainable chemical process was developed through the Fischer esterification of Protobind^®^ lignin, a wheat straw soda lignin, and phloretic acid, a naturally occurring phenolic acid. It aimed at increasing the reactivity of lignin by enhancing the number of unsubstituted phenolic groups via a green and solvent-free chemical pathway. The structural features of the technical and esterified lignins were characterized by complementary spectroscopic techniques, including ^1^H, ^13^C, ^31^P, and two-dimensional analysis. A substantial increase in *p*-hydroxyphenyl units was measured (+64%, corresponding to an increase of +1.3 mmol g^−1^). A full factorial design of the experiment was employed to quantify the impact of critical variables on the conversion yield. The subsequent statistical analysis suggested that the initial molar ratio between the two precursors was the factor predominating the yield of the reaction. Hansen solubility parameters of both the technical and esterified lignins were determined by solubility assays in multiple solvents, evidencing their high solubility in common organic solvents. The esterified lignin demonstrated a better thermal stability as the onset of thermal degradation shifted from 157 to 220 °C, concomitantly to the shift of the glass transition from 92 to 112 °C. In conclusion, the esterified lignin showed potential for being used as sustainable building blocks for composite and thermoset applications.

## 1. Introduction

A recent study from the Joint Research Center of the European Commission of Science revealed that bio-based chemicals represent only 3% of the global production in Europe [1]. It illustrates the strong dependence of the chemical industry on products derived from non-renewable feedstock. The necessity to reduce the dependence on limited petroleum resources has led to the emergence of bio-based alternatives. Lignin, a natural source of polyphenols, has been pointed out as a promising breeding ground of renewable carbon [2]. Every year, around 50 million tons of lignin are isolated as a by-product of the pulp and paper industries. Unmodified lignin meets few commercial applications due to the variation of its chemical and physical properties, closely related to the wood source and the isolation procedure [3]. However, promising recent applications have been reported [4,5,6,7]. The abundant aromatic and aliphatic hydroxyl functionalities contained in the structure of lignin affords a sustainable platform of chemical modification [8,9]. 

Phenolation is the most described approach among the numerous processes employed for the chemical modification of lignin [10,11,12,13,14,15,16]. Phenolation aims at improving lignin reactivity towards formaldehyde, and therefore at contributing to develop new prepolymers for the partial or the complete substitution of petroleum-based phenol in phenol–formaldehyde adhesives [13,14,17], phenolic foam [18] or polybenzoxazine thermoset resins [19]. Phenolation consists in a thermally induced condensation of phenols with the side-chain aliphatic hydroxyl groups of lignin, with the aim to increase the number of *p*-hydroxyphenyl units. Extreme reaction conditions and high amounts of acid catalysts are generally needed to achieve the desired yield of modification. However, such drastic synthetic conditions lead to the fragmentation of lignin through the cleavage of a significant part of ether interunit linkages [10,11,15,16]. Moreover, even if substantial efforts have been done to employ synthons from renewable feedstocks [20,21,22], the synthesis pathway generally suffers from the use of petroleum-based harmful reactants. 

Esterification is one of the oldest chemical reactions to functionalize lignin polyols [23]. Among the different esterification pathways, such as the condensation with acid anhydrides [24,25,26,27] or acyl chlorides [28,29,30,31,32,33], the Fischer esterification appeared as a straightforward strategy to produce esterified lignin [34,35,36,37,38], being more respectful toward the principles of green chemistry. For instance, dimer fatty acids from vegetable oils were successfully grafted onto lignin [34]. Sivasankarapillai et al. reported the synthesis of highly branched lignin-poly(ester-amine) or poly(ester-amine-amide) networks from the condensation of commercially available soda Protobind^®^ lignin with branched carboxylic acid prepolymers [35,36]. More recently, Liu et al. proposed a sustainable strategy fitting most of the green chemistry principles for the esterification of a softwood kraft lignin and an organosolv hardwood lignin [37,38]. Several aliphatic organic acids of different chain lengths were concomitantly used as solvents and reagents. An innovative approach would be to find a sustainable synthetic pathway gathering both the phenolation and esterification of lignin.

In this study, a strategy was developed to esterify lignin with the aim to enhance the number of phenolic groups in a sustainable and environmentally friendly manner. Through a solvent-free Fischer esterification, Protobind^®^ lignin was successfully reacted with phloretic acid, a bio-based phenolic acid originated from the leaves of apple trees [39]. Phloretic acid stands out due to its unique structure, a propionic acid terminated by a 4-hydroxyphenyl group, well-suited to the development of sustainable materials [40]. Advantageously, the oligomerization of phloretic acid is limited due the low reactivity of phenol under Fischer conditions. Protobind^®^ lignin is a technical lignin extracted from wheat straw agro-based residues. A sulfur-free soda pulping process was employed to isolate lignin from crops [41,42]. The structure of the technical lignin differs from native lignin by the chemical modification of ether interunit linkages occurring during the delignification process. Only the aromatic repeating units and methoxyl groups remain similar throughout the isolation procedure. In addition to the high number of aliphatic hydroxyl (‒OH) groups, this sulfur-free technical lignin contains carboxylic acid moieties formed during the pulping process and the three types of aromatic units, namely guaiacyl (G), syringyl (S), and *p*-hydroxyphenyl (H). A design of experiment (DOE) was drawn up to identify and quantify the dependence of pinpointed variables on the yield of conversion of the aliphatic ‒OH into aromatic ‒OH groups. The structural features of the esterified lignin were characterized by several spectroscopic techniques. 2D HSQC NMR was specifically performed to follow the structural modification of lignin substructures upon esterification. To the best of our knowledge, the enhancement of the number of phenolic groups of lignin by a solvent-free esterification with synthons from renewable resources has never been reported so far. The sustainable chemical pathway developed in this work aims to improve lignin reactivity to design alternatives to petroleum-based phenolic compounds.

## 2. Materials and Methods

### 2.1. Materials and Reagents

The different grades of the soda Protobind^®^ lignin were purchased from the lignin company Tanovis AG (Switzerland, lignin > 90%, xylose < 4%). Prior to use, the lignin was dried overnight at T = 50 °C under reduced pressure (P = 10^−2^ mBar). All solvents and chemicals were purchased from Sigma–Aldrich (Overijse, Belgium) and used without further purification. The list of solvents is as follows: acetone (ACE), acetonitrile (ACN), butanone (MEK), chloroform (CHCl_3_), dichloromethane (DCM), diethyl ether (DET), dioxane (DIOX), dimethylformamide (DMF), dimethylsulfoxide (DMSO), ethanol (EtOH), ethyl acetate (ETAC), methanol (MeOH), pyridine (PYR), tetrahydrofuran (THF), toluene (TOL), water (H_2_O), and basic water (NaOH, 1 M).

### 2.2. Esterification of Lignin

The scheme of the reaction is depicted on Scheme 1. In a 250 mL three-neck round bottom flask, dried lignin (brown powder, 2 g), phloretic acid (PA) and para-toluene sulfonic acid (*p*-TSA) were accurately weighed and mixed together by a motorized arm (200 rpm) at room temperature under a small flow of argon. Then, the temperature was gradually raised to T = 140 °C, above the melting temperature of phloretic acid (129 °C). The reaction was monitored under argon atmosphere. A water trap (calcium chloride, CaCl_2_) was used to shift the reaction equilibrium towards the ester formation. At the end of the reaction, the temperature of the reaction medium was cooled down to room temperature. The resulting dark-brown residue was solubilized in 200 mL of acetone to collect the whole product. After the complete evaporation of acetone in open-air, the reaction crude was stirred overnight with 200 mL of diethylether to remove the excess of unreacted PA. Esterified lignin (light-brown powder) was collected after filtration, several washings with diethylether and a final drying step (T = 50 °C overnight under reduced pressure). 

A DOE was carried out following a 2^3^ full factorial design. An experimental matrix was employed to study the effect of critical variables on the reaction yield: the reaction time (**t**)the molar ratio between the carboxylic acid units from PA (n_COOH_PA_) and the lignin aliphatic ‒OH groups determined by ^31^P NMR (n_OHali_P2400_) (**n**)the catalyst loading expressed as weight percentage (wt.%), relative to the initial amount of lignin (**c**)

The extreme levels of each critical variable were selected according to the common values reported in the literature and are tabulated in Table 1 (discrete values). The experimental matrix was generated randomly considering the extreme values of each variable and analyzed with Ellistat software (student license). All the experiments were run.

### 2.3. Equipments and Characterization

Nuclear magnetic resonance (NMR) spectroscopy was recorded using an AVANCE III HD spectrometer (Bruker, Fällanden, Switzerland) equipped with a 5 mm BBO-probe, operating at a proton frequency of 600 MHz. ^31^P NMR analysis was performed following the common procedure reported by Granata and Argyropoulos [43]. 30 mg of dried lignin was accurately weighed in a 1 mL vial and afterwards dissolved in 500 μL of the solvent mixture (A) containing deuterated chloroform (CDCl_3_) and anhydrous pyridine in a 1/1.6 (*v/v*) ratio. The mixture of solvents (A) was also used for the preparation of the relaxation agent and the internal standard solutions (respectively, B and C). Chromium (III) acetylacetonate (14 μmol mL^−1^) and endo-N–hydroxy–5–norbornene–2,3–dicarboximide (108 μmol mL^−1^) were selected as the relaxation reagent and the internal standard compounds, respectively [44]. Phosphitylation was performed by adding 50 μL of 2-chloro-4,4,5,5-tetramethyl-1,3,2-dioxaphospholane (TMDP) just after the addition of solutions A, B and C (respectively, V = 150, 100 and 100 μL). All ^31^P NMR spectra were calibrated on the H_2_O derivatized peak (sharp signal at δ = 132.2 ppm). The choice of the internal standard (IS), the assignment and the spectral integration limits of every peak was made according to the relevant suggestions reported by Balakshin et al. [45]. The ^1^H NMR spectra of lignin samples (15 mg) were recorded in 0.6 mL of DMSO-d_6_. The acquisition parameters were as follows: 25 °C, 12,019 Hz spectral width, 128 scans, 2.7 s acquisition time and 10 s relaxation delay (D1). All the chemical shifts were referenced to the DMSO-d_6_ solvent peak (δ_C_ = 39.5 ppm, energy from hydrogen bonds between molecules (δ_H_) = 2.49 ppm). The ^13^C NMR experiments were adapted from the procedure developed by Capanema et al. [46]. Typically, 16 mmol of Cr(acac)_3_ and 200 mg of lignin were dissolved in 550 μL of DSMO-d_6_. The acquisition parameters were as follows: 25 °C, 36,232 Hz spectral width, 20,000 scans, 1.4 s acquisition time, and 2.5 s relaxation delay. The ^1^H-^13^C 2D HSQC NMR experiment was adapted from the procedure reported by Tran et al. [47]. Typically, 200 mg of lignin were dissolved in 700 μL of DSMO-d_6_. The 2D HSQC spectra were acquired using the standard Bruker pulse sequence “hsqcetgpsp.3”. The acquisition parameters were as follows: 25 °C, 24 scans. F1 dimension: 25,641 Hz spectral width, 5.6 ms acquisition time. F2 dimension: 7212 Hz spectral width, 170 ms acquisition time. The ^1^H -^13^C 2D HMBC NMR experiment was adapted from the procedure reported by Huber et al. [48]. The 2D HMBC spectra were acquired using the standard Bruker pulse sequence “hmbcgplpndqf”. The acquisition parameters were as follows: 25 °C, 16 scans. F1 dimension: 31,646 Hz spectral width, 32.4 ms acquisition time. F2 dimension: 6602 Hz spectral width, 310 ms acquisition time. 

Elemental analysis (CHNS/O measurements) was performed on a Vario MACRO cube (Elementar France SARL, Lyon, France). Samples were put into an oxygen-enriched furnace at 1150 °C, where a combustion process converted carbon to carbon dioxide; hydrogen to water; nitrogen to nitrogen gas/oxides of nitrogen and sulfur to sulfur dioxide. The combustion products were heated separately to the corresponding desorption temperature (T_desorpt._) in order to release the components as follows: CO_2_ (T_desorpt._ = 240 °C), H_2_O (T_desorpt._ = 150 °C) and SO_2_ (T_desorpt._ = 100 °C or 230 °C). 

Fourier transform infrared spectroscopy (FTIR) was performed by using a TENSOR 27 (Bruker, Ettlingen, Germany) instrument in the attenuated total reflection (ATR) mode using a diamond crystal. All spectra were recorded at room temperature in a direct absorbance mode and a frequency range of 4000 to 400 cm^−1^ with 16 scans averaged at a 4 cm^−1^ resolution.

Solubility tests were performed similarly to the description by Sameni et al. with minor modifications [49]. 50 mg of dried lignin samples were mixed with 5 mL of organic solvent and subsequently sonicated for 15 min in a water bath sonicator. The insoluble fraction was filtered on 0.2 μm agilent syringe filters. The lignin samples were weighed before the mixing and after the evaporation of the solvent (one day under the fume hood and overnight under reduced pressure at T = 50 °C, if necessary). The temperature was raised up to 80 °C when high-boiling-point solvents were employed. The soluble fraction was calculated as follows (1):S (wt.%) = 100 × [*m*_i_ − *m*_f_]/*m*_f_(1)
where “*m*_i_” corresponds to the initial mass of the lignin samples and “*m*_f_” the mass of the soluble fraction. The lignin samples were subjected to an experimental estimation of Hansen solubility parameters (HSPs) [50]. The samples were graded as compatible with the solvent if S > 80 wt.% or incompatible with the solvent if S < 80 wt.%. With the help of the generated data set, the HSPs of each lignin samples were calculated using the HSPiP 5th edition, 5.0.13 software (Copyright © 2008–2015 Steven Abbott and Hiroshi Yamamoto).

Gel permeation chromatography (GPC) was performed on a 1260 Infinity II gel permeation chromatograph (Agilent technologies, Craven Arms, United Kingdom) to determine the molecular weight of the lignin samples. The chromatograph was equipped with an integrated IR detector, PLgel 5 mm MIXED-C, PLgel 5 mm MIXED-D columns and a PLgel guard column (Agilent Technologies, USA). THF was used as an eluent with a flow rate of 1.0 mL min^−1^ at 40 °C. Polystyrene standards (Agilent Technologies, *M*p = 162–1500 × 10^3^ g mol^−1^) were used to perform the calibration of the system. The molecular weight is determined on the THF’s soluble fraction part (4 mg mL^−1^).

Differential scanning calorimetry (DSC) thermograms were recorded on the DSC 3+ device (Mettler Toledo, Greifensee, Switzerland) in standard pierced aluminum crucibles (40 μL). The sample was initially subjected to a first heating–cooling ramp from 25 to 150 °C to remove residual traces of moisture and clear the thermal history of the lignin samples (N_2_ atmosphere, flow rate: 40 mL min^−1^, heating rate: 10 °C min^−1^, cooling rate: 20 °C min^−1^). The sample was finally reheated from 25 to 225 °C (heating rate: 10 °C min^−1^). The glass transition temperature (*T*_g_) is defined as the midpoint temperature interval when the baseline shift upon heating (endothermic transition). 

Thermogravimetric analysis (TGA) was performed on the TGA 2 device (Mettler Toledo, Greifensee, Switzerland) in a standard ceramic alumina pan from 25 to 800 °C (N_2_ atmosphere, flow rate of 40 mL min^−1^, heating rate: 10 °C min^−1^). The onset of the degradation temperature and the major degradation temperatures were determined using the derivative of the TGA curves (DTG). The DTG curves were independently calculated considering the Equation (2):Δ*m*/ Δ*T*= [*m*_i_ − *m*_f_]/ [*T*_i_ − *T*_f_] (2)
where “*m*” stands for the relative mass (%) and “*T*” the temperature (°C).

## 3. Results

### 3.1. Structural Characterization of the Esterified Lignin

As phloretic acid melts above 129 °C, its mixture with lignin in the presence of *p*-TSA allows the reaction to be performed in melt at 140 °C, at the condition the lignin is thermally stable at this temperature. Among the different commercial grades proposed by Tanovis AG, Protobind^®^ 2400 technical lignin (P2400) was selected based on this parameter (Appendix A), as the onset of its thermal degradation is not reached before 157 °C. The structural features of the esterified lignin (P2400-PA) were characterized by ^1^H, ^13^C and HSQC NMR. Due to the complex structure of lignin, the three techniques were essential to accurately define the structure of P2400 and to determine the structural changes coming from the esterification. ^31^P was also performed to quantify the rate of esterification. It is worthwhile to note that the esterification reaction was performed solventless. The ^1^H NMR spectra of P2400 and P2400-PA are reported in Figure 1. 

For P2400, the peaks located at δ = 3.44, 3.52, 3.76 and 12.48 ppm correspond to the protons adjacent to an oxygen atom (α-protons adjacent to the aliphatic ‒OH groups and the etherified interunit linkages, H_x_ and ‒O‒Alk, respectively), the methoxyls (‒OCH_3_), and the carboxylic acid (‒COOH) moieties, respectively (Figure 1a). The characteristic aromatic protons of the three phenylpropane units are attributed to the broad signal at δ = 6–7.5 ppm. The success of the esterification reaction was confirmed by the appearance of new peaks at δ = 2.57 and 2.75 ppm corresponding to the methylene protons (2, 3) of the bridge between the ester bond and the *p*-hydroxyphenyl group. In the aromatic area (δ = 6–8 ppm), the chemical modification was reflected by the appearance of two signals (5, 6) corresponding to the aromatic protons of the PA counterpart (δ = 6.69 and 7.02 ppm, respectively). The emergence of a strong peak at δ = 9.21 ppm is consistent with the introduction of phenolic groups (8). In addition to these new peaks, the characteristic α-protons adjacent to the aliphatic ‒OH groups shifted downfield due to the new chemical environment (H_x’_, δ = 4.12 ppm).

Quantitative ^13^C NMR was employed to provide more details about the structure by the identification of quaternary and oxygenated carbon over the spectral range δ = 20–180 ppm (Figure 2). 

The assignment of lignin’s characteristic peaks was based on the comprehensive studies conducted by Capenama and coworkers [46,51]. Only peaks from the etherified interunit linkages and ‒OMe are clearly noticeable for the technical lignin (Figure 2a). The intense and sharp signals in the oxygenated aliphatic carbon range (δ = 58–90 ppm) suggested that the interunit linkages and the aliphatic ‒OH groups of P2400 were chemically modified by the supplier during the isolation procedure [52]. The signal at δ = 56.5 ppm is assigned to the characteristic methoxyl groups (‒OMe). The methylene carbon in α-position of the aliphatic ‒OH groups (C_x_) and etherified interunit linkage (‒O‒Alk) are observed at δ = 60.8 and 70.3 ppm, respectively. The success of the reaction was confirmed by the shifting of the peaks associated to carbon in α-position of the aliphatic ‒OH groups, indicating a modification of the chemical environment (C_x’_ at δ = 63.8 ppm, Figure 2b). Aliphatic carbon atoms associated to the propionic chain of PA appeared at lower chemical shift (C_2_ and C_3_ at δ = 36.1 and 30.0 ppm, respectively). The most significant evidence of the success of the esterification from ^13^C NMR is the emergence of a peak in the aliphatic carboxyl range associated with the ester bond (C_1_, δ = 172.8 ppm). It is noteworthy that the absence of conjugated ester peaks (δ = 166–168 ppm) originated from the esterification of phenol moieties, which underlines the selectivity of the Fisher esterification towards aliphatic ‒OH groups [46]. The aromatic region spans several spectral ranges according to the nature of the carbon, i.e., aromatic methine carbons (δ = 103–125 ppm), aromatic carbon–carbon moieties (non-oxygenated quaternary carbons, δ = 125–141 ppm), and aromatic oxygenated carbons (δ = 141–160 ppm) [53]. The number of methoxyls and aliphatic carboxyl functionalities per aromatic ring was calculated through a semi-quantitative interpretation of the ^13^C NMR spectra. The number of carbons in the aromatic region (δ = 103–160 ppm) was normalized to 600 corresponding to 6 × 100 aromatic units (value expressed per 100 Ar) [51]. The methoxyl content (δ = 58–54 ppm) decreased from 132/100 Ar to 77/100 Ar for P2400 and P2400-PA, respectively. The amount of aliphatic carboxyl functionalities (δ = 168–175 ppm) shifted from 14/100 Ar for P2400 to 42/100 Ar for P2400-PA. The decrease in methoxyl groups through the increment of *p*-hydroxyphenyl units, together with the increase in carboxyl functionalities in the esterified lignin depicts the success of the reaction.

The HSQC spectra of P2400 and P2400-PA are reported in Figure 3 over two regions of interest: the aliphatic side chain (δ_C_/δ_H_: 25–80/2.4−4.4) and the aromatic (δ_C_/δ_H_: 100−155/6.0−8.0) regions. On the figures are also reported the structures and substructures identified in P2400 and P2400-PA. Cross peak signals listed in Appendix A were assigned with respect to comprehensive databases reported for wheat straw lignins [54,55,56,57]. In P2400, the most intense correlation was identified as methoxyl (‒OMe) groups (δ_C_/δ_H_: 56.5/3.76, Figure 3a). Although the β-aryl ether (β-O-4’) stands for the most occurrent interunit linkages in wheat straw lignin [54], only the characteristic cross peak of the methylene in α-position of the lignin aliphatic ‒OH groups is observed (A_x_, δ_C_/δ_H_: 60.8/3.52). This observation is in line with the chemical modification operated on interunit linkages during the extraction process (Figure 2a). Minor amounts of resinol (β-β’) substructures were also detected (C_β_ and C_γ_). Weak cross peak signals located at δ_C_/ δ_H_ = 73.1/3.13, 74.6/3.33, and 75.9/3.58 were assigned to the presence of xylan moieties (X_2_, X_3_ and X_4_), as previously reported for soda lignin [54,56,57]. The aromatic region of P2400 highlighted the presence of the three types of phenylpropane units through syringyl (S_2_ and S_6_), guaiacyl (G_2_, G_5_ and G_6_), and *p*-hydroxyphenyl (H_2,6_ and H_3,5_) cross peak signals (Figure 3c). Other minor substructures were distinguishable, including signals corresponding to α-oxidized esterified syringyl units (S’_2,6_), ferulate (Fa) and *p*-coumarate (Pc) derivatives (Appendix A) [54,55,57]. The success of the esterification was confirmed by signal modifications in the aliphatic side chain region of P2400-PA (Figure 3b). The signal corresponding to the α-methylene groups (A_x_) was replaced by a new cross peak associated to the esterified substructures (B_x’_, δ_C_/δ_H_: 63.8/4.12). The presence of this new peak confirms the involvement of the aliphatic ‒OH groups in the esterification to form the new structure noted B on Figure 3. Two new correlations appear between δ_H_ = 2–3 ppm, corresponding to the methylene bridge of PA counterparts (B_2_ and B_3_). Resinol (C) and xylan (X) substructures were not affected by the esterification process. In the aromatic region, only peaks originated from the esterification were identified (Figure 3d). The esterification of P2400 causes the appearance of aromatic correlations associated to the new phenolic units from PA counterparts (B_5,6_). Signals of α-oxidized esterified syringyl units, ferulate and *p*-coumarate derivatives disappeared. Moreover, HMBC experiments were also performed on P2400-PA to substantiate the connectivity of ester moieties to the lignin backbone (Appendix A) [48]. The spectral window region was chosen to show the correlations associated with the esterified substructure B (δ_C_/δ_H_: 125−180/2.4−4.4). The correlation of the carboxyl carbon (C_1_, δ = 172.8 ppm) with the methylene bridge of PA (H_3_ and H_2_, δ = 2.75 and 2.57 ppm, respectively) and the α-methylene protons (H_x’_, δ = 4.12 ppm) reflects the success of the esterification.

Finally, the carboxylic, aromatic and aliphatic ‒OH groups of lignin were identified and quantified by ^31^P NMR experiments [43]. The interval of the different types of lignin ‒OH groups were assigned as follows: aliphatic ‒OH (δ_OHali_ = 146–150 ppm), 5-substituted aromatic ‒OH (S and G condensed units, δ_5-susbt._ = 141–144.5 ppm), G non-condensed aromatic ‒OH (δ_Gnc_ = 138.5–141 ppm), *p*-hydroxyphenyl aromatic ‒OH (δ_H_ = 137–138.5 ppm), and carboxyl groups (δ_COOH_ = 134–136 ppm) (Appendix A) [45]. A magnification of the ^31^P NMR spectra of P2400 and P2400-PA between δ = 134–149 ppm is reported in Figure 4.

The unusual sharp signal observed in the aliphatic region of P2400 ([OH]_ali_, δ = 147.1 ppm) tends to confirm the modification of the aliphatic ‒OH groups, as firstly reported by Azhavi et al. [52]. A significant drop of the aliphatic ‒OH groups concomitantly to the substantial increase in *p*-hydroxyphenyl units ([H], δ = 137.4 ppm) were observed by comparing both ^31^P spectra. The results of the quantitative determination of lignin ‒OH groups through ^31^P NMR analysis are gathered in Table 2. The amount of lignin functionalities are expressed in mmol of functional group per gram of lignin dry matter (mmol g^−1^). The amount of [COOH] units for P2400-PA is nearly equivalent to the initial amount in P2400 (~0.80 mmol g^−1^). As regards to the aromatic units, only the content in [H] units considerably increased for P2400-PA stemming from the chemical modification ([H] = 1.81 mmol g^−1^). This sustainable esterification process led to an increase of 1.3 mmol g^−1^ in the amount of [H] units. For the sake of comparison, the phenolation pathway applied on a soda lignin contributes to an increase of 0.9 mmol g^−1^ for this type of unit [16]. The lower amount of [5-subst.] and [G_nc_] tends to indicate that thermally induced side-reactions occurred during the esterification. The reaction also led to a significant decrease in the overall amount of lignin aliphatic ‒OH groups from 2.00 mmol g^−1^ for P2400 to 0.14 mmol g^−1^ for P2400-PA. However, the success of the reaction cannot just be followed by considering the decrease in the amount of aliphatic ‒OH groups ([OH]_ali_). Indeed, the high temperature of the reaction is also suitable for lignin self-condensation, depolymerization and repolymerization, which could consume the amount of [OH]_ali_ [58,59,60]. Consequently, the yield of the reaction must be determined by considering the increase in [OH]_aro_, and more specifically the increase in [H] units (Table 2, column 5).

FTIR and elemental analysis provided complementary insights on the structural features of the esterified lignin. For the technical lignin, the weak absorption band at ν = 1705 cm^−1^ corresponds to the conjugated carboxylic acid stretching of the ferulate and the *p*-coumarate derivatives (Appendix A) [61,62]. Although most of the absorption bands of P2400-PA overlapped with the pattern of P2400, a clear shift was observed in the carboxyl area corresponding to the ester stretching (ν = 1728 cm^−1^), confirming the success of the esterification. The elemental analysis (Table 3) indicated that a relatively low content of nitrogen and sulfur originated from the extraction process were measured (~0.6%) [63,64]. In P2400-PA, the carbon content increased at the expense of the oxygen content, in accordance with the chemical modification (67.20% and 25.89%, respectively). 

### 3.2. Optimization of the Fischer Esterification of Protobind^®^ Lignin 

A design of experiment methodology (DOE) was applied to identify and quantify the dependence of the reaction parameters on the solvent-free Fischer esterification of P2400 with PA. The temperature of the reaction was fixed at T = 140 °C, i.e., between the melting temperature of PA (T_m_ = 129 °C) and the onset temperature of thermal degradation of P2400 (T_onset_ = 157 °C). The reaction time (t), the initial molar reactant ratio (n) and the catalyst loading (c) were identified as the critical variables of the esterification (Table 1). A two-level three factors full factorial experiment (2^3^) was built considering all the possible combinations of the extreme levels of the critical variables. Therefore, a series of eight experiments were completed (Table 4). Thus, the success of the esterification was defined by following the increase in the amount of [H] units. This designation encompasses the original *p*-hydroxyphenyl units of lignin and the 4-hydroxyphenyl groups from PA counterparts. It is worthwhile to indicate that unreacted PA was removed by washing several times with DET. The response associated to the DOE corresponds to the increase in the amount [H] units (determined by ^31^P NMR spectroscopy) for the esterified lignin. The response was calculated according to Equation (3): Y (%) = 100 × [X_*H_P2400-PA*_ − X_*H_P2400*_]/X_*OHali_P2400*_(3)
where “X*_H_P2400-PA_*” (mmol g^−1^) stands for the amount of [H] units in the esterified lignin (EL), “X*_H_P2400_*” (mmol g^−1^) for the initial amount of [H] units in P2400 and “X*_OHali_P2400_*” (mmol g^−1^) for the initial amount of [OH]_ali_ groups in P2400. 

The conversion of the aliphatic ‒OH groups is relatively high for all esterified lignin (≥70%). However, the evolution of the [H] units differs, since it is independent from lignin self-condensation, depolymerization and repolymerization. The following observations can be highlighted:when a stoichiometric amount of reactants is used (n = 1; Table 4, rows 1,2, 5 and 6), the increase in the amount of [H] units does not exceed 28%,when an excess of PA is used (n = 5; Table 3, rows 3, 4, 7, 8), higher conversion yields of aliphatic ‒OH groups (~90%) and amount of [H] units are reached (~50% and 65 % for t = 12 and 48 h, respectively),for n = 1, the amount of [H] units is equivalent to ~ 0.9 and ~1.1 mmol g^−1^ (c = 0.5 and 2.5 wt.%, respectively) independent of the reaction time,for an excess of PA (n = 5), this amount is equivalent to ~1.6 and 1.8 mmol g^−1^ (t = 12 and 48, respectively) regardless of the catalyst loading.

Crosschecking these results indicates the amount of catalyst affects the increase in [H] units only when PA is introduced in stoichiometric proportion (n = 1). The highest increase in [H] units is obtained when an excess of PA is used for a longer reaction time (Table 4, rows 4 and 8). The catalyst loading does not significantly affect the increase in [H] units under these experimental conditions ([H] = 1.78 and 1.81 mmol g^−1^ for c = 0.5 and 2.5 wt.%, respectively). Under these optimal reaction conditions (n = 5, t = 48 h), most of the aliphatic ‒OH groups were converted into ester counterparts (93 %). In the context of this DOE, the initial molar ratio (n) appears as the main critical variables affecting the yield of the esterification. To confirm this result, the reaction time and the catalyst loading were both increased (from 12 to 48 h and from 0.5 to 2.5 wt.%, respectively). The results show it poorly enhanced the yield of the esterification.

A complete statistical analysis was carried out based on the experimental results using Ellistat software (95% confidence interval). The normalized impact of the individual critical variables (t, n and c) and the first order interaction effect of the temperature with the initial molar ratio (I_tn_), the temperature with the catalyst loading (I_tc_), and the initial molar ratio with the catalyst loading (I_nc,_), were evaluated in variance analysis using a Pareto chart (Figure 5). 

The major contribution comes from the initial molar ratio factor (71.1%). Even if the time of the reaction stands for the less important factor (<2%), its interaction with the initial molar ratio contributes as much as the individual catalyst loading factor (~10%).

A linear predictive model was elaborated considering the most significative factors determined from the Pareto chart (contribution > 5%, Equation (4)):Y = 7.36 + 8.18*n + 4.66*c + 0.06*I_tn_ + 0.91*I_nc_(4)
where “Y” is the response (increase in [H] units), “n” the initial molar ratio, “c” the catalyst loading, “I_tn_“ the first order interaction effect of the temperature with the initial molar ratio and “I_nc_“ the first order interaction effect of the initial molar ratio with the catalyst loading. The predictive model enables one to predict with good accuracy the increase in [H] units in the experimental field (Appendix A, R^2^ = 0.994). The results coming from the analysis of variance (ANOVA) reported in Table 5 indicate that most of the critical variables and their first order interaction effects considered for the elaboration of the predictive model were significant (degree of significance: p < 0.05, Appendix A, raw 2). The initial molar ratio (n) was identified as the factor with the highest probability to influence the response (p = 0.002) (Appendix A, column 4, row 2). Indeed, the weaker the p, the higher the probability that the factor affects the result. An additional experiment was carried out to assess the reliability of this predictive model by fixing the critical variables at the center of the DOE (t:n:c; 0:0:0) corresponding to a time of reaction of t = 30 h with an initial molar ratio of n = 3 and a catalyst loading of c = 1.5 wt.% (EL-9). This experiment confirmed the robustness of the model by a close agreement between the experimental and the model-predicted values (respectively, 43.5 and 48.5 %; details of the calculations reported in the Appendix A). In addition, to assess the pairwise effect of the critical variables, response surfaces (3D-visualization) stand for another efficient tool to estimate the response inside the experimental field (Appendix A). In conclusion, the experimental and statistical analysis shows that the initial molar ratio between the ‒COOH of phloretic acid and the lignin aliphatic ‒OH groups is the most significant critical variable affecting the yield of the esterification, as generally reported with Fischer-like esterifications [37,38]. It is worth indicating that an excess of PA helps the mixing of the reactants since the viscosity of the reaction medium is substantially decreased (melt-like condition). It explains, together with kinetic considerations, why the initial molar ratio is the most significant variable in the case of this synthesis. Finally, the DOE leads to the conclusion that the most suitable conditions are an excess of PA over P2400 (n = 5), an extended time of reaction (t = 48 h) and a higher catalyst loading (c = 2.5 wt.%).

### 3.3. Physicochemical Properties of Esterified Lignin

The solubilities of P2400 and P2400-PA were assessed in several organic solvents covering a wide range of polarity (Figure 6, Appendix A) [49]. Lignin samples were subjected to an experimental estimation of Hansen solubility parameters (HSPs) [50]. To this aim, 50 mg of lignin were immersed in 50 mL of solvent. The percentage of solubility was determined by weighting the insoluble part after proper drying under reduced pressure and heating, if necessary, to remove the solvent. P2400 was highly soluble in most of the solvents. While less than 5 wt.% of P2400 was soluble in DET or TOL, the highest solubilities were reached in ACE, MeOH, DMF, PYR, DMSO, THF and DIOX (S > 95 wt.%). The determination of the solubility of P2400 in the different solvents allows the estimation of its HSPs by using the Hansen solubility parameters in practice software (HSPiP). Estimated values of HSPs (δ_D_ = energy from dispersion forces between molecules, δ_P_ = energy from the dipolar intermolecular forces between molecules, δ_H_ = energy from hydrogen bonds between molecules) are reported in Table 5. A high value of δ_H_ seems to be the most important parameter to trigger the solubility of P2400 due to the high amount of –OH groups (Table 5). This solubility map comforts us in the solvent’s choice for the purification (DET), NMR analysis (PYR for ^31^P, DMSO for ^1^H, ^13^C and 2D spectra), and GPC analysis (THF). It also explains why the reaction depends so much on the initial molar ratio. Indeed, the similarity of the HSPs of PA (δ_D_/δ_P_/δ_H_ = 19.5/7.6/16.1 MPa^1/2^, estimated from the software HSPiP) and P2400 (Table 5, columns 2, 3 and 4) emphasizes why the esterification reaction works in melt conditions, as molten PA could act as the “solvent” of the reaction. The overall solubility of P2400-PA was tightly like P2400. Esterified lignin was fully insoluble in water (neutral or acidic condition). The solubility of P2400-PA in alcohol solvents (EtOH and MeOH) considerably decreased as a result of the chemical modification of the aliphatic ‒OH groups (S = 91 and 96 wt.% for P2400, and S = 41 and 36 wt.% for P2400-PA, respectively). Inversely, the increase in the solubility in ester-containing solvents (ETAC and MEK) originates from the creation of ester bonds. The high solubility of the esterified lignin in a wide range of solvents is a key factor for subsequent engineering applications, for instance solution-state chemical modifications, composites processing, etc.

The molecular weight of P2400 and P2400-PA was determined by GPC-SEC experiments (Appendix A). The number average molecular weight (*M*_n_), the weight average molecular weight (*M*_w_), and the dispersity (*Đ*) are reported in Table 5. The low molecular weight of P2400 is in the same range as the Protobind^®^ lignin samples (*M*_n_ = 575 g mol^−1^) [57,63]. The esterification of P2400 with PA is characterized by an increase of the molecular weight from 575 to 1570 g.mol^−1^. The narrower dispersity of P2400-PA as compared to P2400 (5.4 and 7.5, respectively) could be explained by the extraction of low molecular weight compounds during the extraction process. The higher molecular weight of P2400-PA compared to P2400 is in good accordance with previous studies conducted on the esterification of lignin [32,33,37]. 

Finally, the thermal behaviors of P2400 and P2400-PA were investigated by thermogravimetric analysis (TGA) and differential scanning calorimetry (DSC). The TGA and first derivative (DTG) curves are displayed in Figure 7. Below 100 °C, a low weight loss was caused by the gradual evaporation of water (moisture content, <2%). The technical lignin follows a three-stage degradation pattern, starting from the dehydration of aliphatic ‒OH groups, followed by interunit linkages cleavage, and terminated with the decomposition of the polymer backbone (Appendix A). In Table 6 the onset temperature of degradation (*T*_onset_, determined from the first derivative), the maximum temperature of the main degradation stage (*T*_max_), and the char residue of P2400 and P2400-PA are summarized. Lignin ester derivatives exhibited improved thermal stability as the *T*_onset_ shifted from 157 to 220 °C for P2400 and P2400-PA, respectively. For the esterified lignin, a two-stage degradation pattern was observed [32]. The *T*_max_ of the main degradation stage was also slightly higher for the P2400-PA than for P2400 (*T*_max_ = 367 and 359 °C, respectively). The higher value of residual char for P2400-PA (37.6%) was related to the higher carbon content (elemental analysis, Table 2), as well as to the increased number of phenolic rings. It also led to a small increase of the low oxygen index (LOI), from 31.1% to 32.5%, indicative of a slightly improved fire-retardant behavior [65]. The glass transition (*T*_g_) of each material was measured on the second heating ramp from DSC measurement, evidencing the *T*_g_ of P2400 was increased from 92 to 112 °C after its esterification with PA (Table 6, column 6; Appendix A). 

## 4. Conclusions

This work details how the reactivity of a soda Protobind^®^ 2400 lignin can be enhanced by its solvent-free reaction with phloretic acid, fitting many of the green chemistry principles. A two-level full factorial DOE was carried out to spot the variables of interest, and consequently to determine the optimal experimental conditions and to elaborate a reliable predictive model. The excess of phloretic acid appeared to be the most effective parameter monitoring the yield of the esterification. The esterification of Protobind^®^ lignin with phloretic acid contributed to considerably increase the amount of *p*-hydroxyphenyl units (from 0.53 to 1.81 mmol g^−1^). The structural features of the esterified lignin were substantiated by complementary NMR techniques (^31^P, ^1^H, ^13^C, 2D HSQC and HMBC), FTIR and elemental analysis. Esterification stood for the main reaction occurring during the chemical process with a yield of reaction reaching 64 % (increase in [H] units). Minor side-reactions, such as self-condensation, were also detected. Solubility assays in various types of solvents were performed to determine the Hansen solubility parameters and to draw the solubility mapping of P2400 and its esterified form with phloretic acid. It highlighted a high solubility of both P2400 and P2400-PA in solvents of high δ_H_, an important knowledge for practical applications and future research perspectives. Finally, the thermal properties of the esterified lignin were assessed by TGA and DSC. The onset of the thermal degradation shifted from 157 to 220 °C, concomitantly with the enhancement of the *T*_g_ from 92 to 112 °C, respectively, for P2400 and P2400-PA. To conclude, esterified lignin containing large amounts of phenolic units embodies a sustainable precursor to produce biobased and high-value-added materials of higher thermal stability. The promising chemical process developed in this work aims to magnify the reactivity of lignin towards a panel of industrial applications such as wood-based composites or thermoset. 

## Data Availability

Not applicable.

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
