# Peer review of "Sustainable Esterification of a Soda Lignin with Phloretic Acid"

_polymers, 2021, doi:10.3390/polym13040637_

Round 1
Reviewer 1 Report
The esterification process of lignin presented by the authors is well presented and experimental part is of high quality. Proposed procedure provides a simple and straightforward way to modify lignin and to increase the number of Ar-OH groups at the expense of Alk-OH groups.
Only minor suggestions are proposed:
- Better description of the lignin will ease the use of proposed method with other lignin as a starting material
- Please specify the origin of the lignin (is it a hardwood or a mix of hard and soft wood)
- Lignin presented in the study has residue carbohydrates, please specify the purity of the lignin. What is Klasson lignin value for your lignin sample?
- Page 3 “the result dark-brown pulp…” please replace pulp with residue a synonym you like. Pulp is usually used for carbohydrates residue
- For GPC please specify columns and PS standard Mw range/values.
- Please replace in the text PDI.
IUPAC has deprecated the use of the term polydispersity index, having replaced it with the term dispersity, represented by the symbol Đ.
Author Response
The authors are warmly thankful to the reviewers for their kind and very constructive comments. They will definitively increase the quality of the paper. We considered very attentively the comments and a point-by-point answer is given below.
All the modifications done in the revised version of the manuscript are highlighted in yellow.
Finally, the template of the document was converted to the most recent template of Polymers (2021).
- Reviewer 1:
The esterification process of lignin presented by the authors is well presented and experimental part is of high quality. Proposed procedure provides a simple and straightforward way to modify lignin and to increase the number of Ar-OH groups at the expense of Alk-OH groups
We would like to thank the Reviewer for his/her pleasant appreciation of the work we proposed.
- Better description of the lignin will ease the use of proposed method with other lignin as a starting material
The reviewer is right. A better description of the lignin has been added in the revised manuscript. “Protobind® lignin is technical lignin extracted from wheat straw agro-based residues. A sulfur-free soda pulping process was employed to isolate lignin from crops. The structure of the technical lignin differs from native lignin by the chemical modification of ether interunit linkages occurring during the delignification process. Only the aromatic repeating units and methoxyl groups remain similar throughout the isolation procedure.”
Please specify the origin of the lignin (is it a hardwood or a mix of hard and soft wood)
P2400 is not extracted from wood but annual crops (straw, sugarcane bagasse, and flax). The description of the origin of lignin was added to the introduction section concomitantly with the description of lignin reported above.
- Lignin presented in the study has residue carbohydrates, please specify the purity of the lignin. What is Klasson lignin value for your lignin sample?
The purity of the lignin was specified in the Material and Methods section (page 2) (lignin> 90 %, xylose< 4 %). For P2400, the experimental Klasson lignin value was found at 88 ± 1 %, in close agreement with the lignin content given by the supplier (DOI: 10.1016/j.indcrop.2013.10.014).
- Page 3 “the result dark-brown pulp…” please replace pulp with residue a synonym you like. Pulp is usually used for carbohydrates residue
The Reviewer is right. This is now corrected in the revised text.
- For GPC please specify columns and PS standard Mw range/values.
As suggested by the Reviewer, we added more details to the GPC analysis in the revised manuscript.
- Please replace in the text PDI. IUPAC has deprecated the use of the term polydispersity index, having replaced it with the term dispersity, represented by the symbol Đ.
We would like to thank the reviewer to notice this mistake. It is corrected in the revised version of the manuscript.

Reviewer 2 Report
I read carefully the review article entitled ‘Insights on the structural features of a soda lignin and its optimized solvent-free Fischer esterification
with phloretic acid’. The concept of the manuscript fits and is suitable to publish in Polymers Journal. This manuscript is generally well written and clearly presented however some comments should consider to improve the quality of the manuscript
1) Title should be modified in a precise way.
2) Author should avoid using abbreviations in the abstract section. In abstract authors should discuss the importance of research work in one or two sentences.
3) In the manuscript many abbreviations are used so add all abbreviations and their full form after the abstract section would be better.
4) Provide a nice graphical abstract representing the overview of the MS with key highlights.
5) In the introduction section, write the novelty of the work and the problem statement clearly. Avoid clusters of references, give details for example in the introduction first para authors put 4-10 references for one sentence. Authors should discuss some recent applications of lignin Viz. NPs synthesis and biopolymers production pl refer an cite International journal of biological macromolecules 128, 391-40, 2019; Bioresource Technology Volume 325, April 2021, 124685. The detailed discussion about the novelty, significance of your research work and research gap relative to the literature is essential.
6) In figure and table always give full form of abbreviation. In addition, the figure and table caption give all details.
7) Statistical analysis of the results should be provided in the materials and methods section. It's important for all experimental work Report these values in the results and discussion.
8) Surprisingly very little discussion of results with the previous results of literature need substantial discussion at the revision stage. Use recent references from the year 2018-2020.
9) Write the practical applications and future research perspectives and challenges by adding a new section before conclusions.
10) Also include What are the limitations to use this methodology for commercial application?
11) The conclusion of the study is not discussed with the specific output obtained from the study, it could be modified with precise outcomes with a take home message.
12) English and grammar mistakes are present. The author should check the manuscript by native English Speaker to improve the quality of the manuscript.
Author Response
The authors are warmly thankful to the reviewers for their kind and very constructive comments. They will definitively increase the quality of the paper. We considered very attentively the comments and a point-by-point answer is given below.
All the modifications done in the revised version of the manuscript are highlighted in yellow.
Finally, the template of the document was converted to the most recent template of Polymers (2021).
- Reviewer 2:
I read carefully the review article entitled ‘Insights on the structural features of a soda lignin and its optimized solvent-free Fischer esterification with phloretic acid’. The concept of the manuscript fits and is suitable to publish in Polymers Journal. This manuscript is generally well written and clearly presented however some comments should consider to improve the quality of the manuscript.
- Title should be modified in a precise way.
The title was modified in a more precise way: “Sustainable esterification of a soda lignin with phloretic acid”
- Author should avoid using abbreviations in the abstract section. In abstract authors should discuss the importance of research work in one or two sentences.
The reviewer is right. Abbreviations were removed from the abstract. The acronym “NMR” was replaced by “spectroscopic”, while “HSQC and HMBC” simplified to “two-dimensional”. The importance of the research work was emphasized by adding the following sentence ” In conclusion, the esterified lignin showed the potential of being used as sustainable building blocks for composite and thermoset applications.”
- In the manuscript many abbreviations are used so add all abbreviations and their full form after the abstract section would be better.
As suggested by the reviewer, an abbreviation list was added at the end of the manuscript.
- Provide a nice graphical abstract representing the overview of the MS with key highlights.
The graphical abstract was added in the submission process.
- In the introduction section, write the novelty of the work and the problem statement clearly. Avoid clusters of references, give details for example in the introduction first para authors put 4-10 references for one sentence. Authors should discuss some recent applications of lignin Viz. NPs synthesis and biopolymers production pl refer an cite “International journal of biological macromolecules 128, 391-40, 2019”; “Bioresource Technology Volume 325, April 2021, 124685”. The detailed discussion about the novelty, significance of your research work and research gap relative to the literature is essential.
As proposed by the Reviewer, the novelty of the work and the problem statement have been emphasized by adding the following sentences: “An innovative approach would be to find a sustainable synthetic pathway gathering both the phenolation and esterification of lignin” & “The sustainable chemical pathway developed in this work aims to improve lignin reactivity to design alternatives to petroleum-based phenolic compounds” on page 2.
As suggested by the reviewer, clusters of references were removed. The manuscript is referenced by 62 documents, specifically selected in relation to the topic covered by the manuscript. A detailed discussion of the novelty and significance of the work is given all along with the manuscript, which is aiming at filling a research gap about the green phenolation of lignin, a topic never been reported up to now.
Finally, as requested, the two references have been added.
- In figure and table always give full form of abbreviation. In addition, the figure and table caption give all details.
We would like to express our disagreement with the Reviewer. For the sake of clarity and consistency, it is more convenient for the lecturer to read the name of the sample in the abbreviated form (P2400) rather than the full form (ProtobindÒ 2400 lignin). However, to help the clarity, a list of abbreviations was added to the manuscript as suggested by the Reviewer.
- Statistical analysis of the results should be provided in the materials and methods section. It's important for all experimental work Report these values in the results and discussion.
We would like to apologize if we misunderstand the comment of the Reviewer comment is wrong. The way the statistical analysis was provided in the manuscript followed the traditional way it is reported in the literature, for instance in 10.1016/j.eurpolymj.2015.03.029 and 10.1039/D0GC01234C.
- Surprisingly very little discussion of results with the previous results of literature need substantial discussion at the revision stage. Use recent references from the year 2018-2020.
It is correct there is a little discussion of the results compared to the literature. The literature concerning phenolated lignin is generally describing synthetic pathways and the characterization of the structural features. However, the thermal properties, i.e. thermal stability or Tg, are not reported to the best of our knowledge. The only possible discussion is concerning the amount of the number of [H] units. Thus, to follow the Reviewer's recommendation, the following sentence was added to the manuscript.: “This sustainable esterification process led to an increase of 1.3 mmol.g-1 in the amount of [H] units. For the sake of comparison, the phenolation pathway applied on soda lignin contributes to an increase of 0.9 mmol.g-1 for this type of unit”.
- Write the practical applications and future research perspectives and challenges by adding a new section before conclusions.
In the conclusion part, the following sentence was added to fit the request of the Reviewer: “Esterified lignin containing large amounts of phenolic units embodies a sustainable precursor to produce biobased and high-value-added materials of higher thermal stability. The promising chemical process developed in this work aims to magnify the reactivity of lignin towards a panel of industrial applications such as wood-based composites or thermoset”
- Also include What are the limitations to use this methodology for commercial application?
This study reaches a technology readiness level (TRL) of 2. It is very far from the TRL 8-10 corresponding to the levels needed before considering any commercial applications. In the context of this publication, we do not consider it is relevant to detail the limitations of the methodology for commercial applications.
- The conclusion of the study is not discussed with the specific output obtained from the study, it could be modified with precise outcomes with a take home message.
The content of this request was already included in the first version of the manuscript. In the conclusion, the following sentence is written:” Esterified lignin containing large amounts of phenolic units embodies a sustainable precursor to produce biobased and high-value-added materials of higher thermal stability. The promising chemical process developed in this work aims to magnify the reactivity of lignin towards a panel of industrial applications such as wood-based composites or thermoset.”
- English and grammar mistakes are present. The author should check the manuscript by native English Speaker to improve the quality of the manuscript.
Following the Reviewer's suggestion, a careful read of the manuscript was done to improve its quality.
